# Small Tweaks, Major Changes: Post-Translational Modifications That Occur within M2 Macrophages in the Tumor Microenvironment

**DOI:** 10.3390/cancers14225532

**Published:** 2022-11-10

**Authors:** Shutao Zheng, Yan Liang, Yiyi Tan, Lu Li, Qing Liu, Tao Liu, Xiaomei Lu

**Affiliations:** 1State Key Laboratory of Pathogenesis, Prevention and Treatment of High Incidence Diseases in Central Asia, Clinical Medical Research Institute, The First Affiliated Hospital of Xinjiang Medical University, Urumqi 830054, China; 2Department of Pathology, Basic Medicine College, Xinjiang Medical University, Urumqi 830017, China; 3Department of Clinical Laboratory, First Affiliated Hospital of Xinjiang Medical University, Urumqi 830054, China

**Keywords:** M2 macrophages, post-translational modification (PTM), tumor microenvironment (TME)

## Abstract

**Simple Summary:**

Featuring functional diversity and plasticity, M2 macrophages have been typecasting as villain player in tumor microenvironment, propelling the growth and metastases of tumor cells and being closely related with shorter survival. Good things elude them. Why are they so bad? Trying to understand and analyze the behavior of M2 macrophages, we looked at M2 macrophages in a new perspective that is from post-translational modification (PTM) angle, proposing that it is PTMs that occur in M2 macrophages that could endow M2 macrophages with being diverse and plastic in tumor microenvironment. Given this, this review summarizes the advancement of PTMs in M2 macrophages in tumor microenvironment, to gain better understanding of M2 macrophages.

**Abstract:**

The majority of proteins are subjected to post-translational modifications (PTMs), regardless of whether they occur in or after biosynthesis of the protein. Capable of altering the physical and chemical properties and functions of proteins, PTMs are thus crucial. By fostering the proliferation, migration, and invasion of cancer cells with which they communicate in the tumor microenvironment (TME), M2 macrophages have emerged as key cellular players in the TME. Furthermore, growing evidence illustrates that PTMs can occur in M2 macrophages as well, possibly participating in molding the multifaceted characteristics and physiological behaviors in the TME. Hence, there is a need to review the PTMs that have been reported to occur within M2 macrophages. Although there are several reviews available regarding the roles of M2 macrophages, the majority of these reviews overlooked PTMs occurring within M2 macrophages. Considering this, in this review, we provide a review focusing on the advancement of PTMs that have been reported to take place within M2 macrophages, mainly in the TME, to better understand the performance of M2 macrophages in the tumor microenvironment. Incidentally, we also briefly cover the advances in developing inhibitors that target PTMs and the application of artificial intelligence (AI) in the prediction and analysis of PTMs at the end of the review.

## 1. Introduction

As members of the innate immune system, macrophages play pivotal roles in immunity, tissue repair, metabolic homeostasis, and host development [1]. Shaped by different tissue microenvironments, macrophages take on different roles [2]. For example, under physiological conditions, macrophages can change their functional and morphological characteristics and respond to their tissue microenvironments and give birth to resident subpopulations, for example, Kupffer cells in liver [3], alveolar macrophages in lungs [4], microglia in brain [5], and lamina propria macrophages in gut [6], while in the presence of pathological stimuli, such as pathogens or tumor cells, macrophages assume antimicrobe or antitumor roles. Functional diversity, therefore, is one of the key features of macrophages. This functional plasticity can be conditioned by the TME in which they reside. It has been well appreciated that the functional diversity of macrophages can be simplified into two extreme phenotypes, namely M1 and M2 macrophages, with M1 being inflammatory and M2 being anti-inflammatory [7,8]. Many studies have experimentally shown that M2 macrophages play key roles in the TME by assisting tumor cells in accelerating their proliferation and propelling metastases of cancer cells with which they interact. Despite these accounts, the underlying mechanism remains far from being determined. Here, in this review, we will summarize and critically analyze the different types of PTMs reported to have occurred within M2 macrophages in the TME, with the purpose of deepening our understanding of M2 macrophages in the TME. The published literature that concerns M2 macrophages but is unrelated to PTMs in M2 macrophages will be largely disregarded and omitted from this review. However, when reviewing the scarce evidence concerning the PTMs we intended to describe in M2 macrophages, the scope of literature inclusion must be expanded to include tumor-associated macrophages (TAMs) in order to be complete.

## 2. The Origin of M2 Macrophages

Distributed in all tissues and organs of the human body, monocytes and their derived macrophages [9,10] are a predominant component of the host in response to tumors. Influencing tumor development in both positive and negative fashions, macrophages are key players in these interactions [11]. Although their modulation is less well understood and determines the outcome of the relationship between host and tumor, many tumor cells can recruit immature myelomonocytic cells and use them to block their differentiation, eventually inducing peripheral tolerance [12]. In addition, tumor cells can also take advantage of macrophages to foster proliferation, migration, and invasion within their TME [11]. Pioneering observations [13,14,15,16] have strongly suggested that lymphocytes can determine whether macrophages are activated. Subsequent evidence regarding lymphocyte–macrophage communication was provided by Herscowitz HB and coworkers [14]. Based on their distinctive functions, T lymphocytes can be classified into T helper 1(Th1) and T helper 2(Th2) cells [15,16]. Th1 lymphocytes elicit the so-called Th1 immune response by secreting interferon gamma (IFN), interleukin-2 (IL-2), or tumor necrosis factor beta (TNF). Likewise, theTh2 immune response was defined by secretion of interleukins IL-4/5/6/10 and IL-13. Original investigations conducted by Stein M et al. [17], Mills CD et al. [18], and Munder M et al. [19] clearly demonstrated that macrophages will take on distinctive immune functions that hinge on Th1 or Th2 immune responses. Induced by IL-4 and IL-10, macrophages were found to be activated, producing inflammatory cytokines and nitric oxide, enabling pathogen killing and causing inflammation [20]. In contrast, inflammatory cytokines and nitric oxide production can be diminished as long as there is a Th2 immune response [20]. Thus, Th1-induced macrophage activation has been termed M1, which is inflammatory; by analogy, Th2cytokine-induced macrophage activation has been termed M2 [18]. This is where M2 macrophages come into play.

## 3. Differences between M2 Macrophages and TAMs

With regards to the two terms, M2 macrophages and tumor-associated macrophages (TAMs), no confusion should result from their usages, which have subtle differences, such that some may have difficulty distinguishing them. Highly resembling M2 macrophages in terms of function, TAMs in effect include M1 macrophages and M2 macrophages that are functionally much like myeloid-derived suppressor cells (MDSCs) and express various tumor-promoting and immune suppressive factors, thereby fostering tumor proliferation [21] and conferring resistance to being killed by immune mechanisms [22], chemotherapy [23,24], and radiotherapy [25,26]. Therefore, in this sense, TAMs are most similar to M2 macrophages; these terms can sometimes be used interchangeably, unless otherwise specified. However, in light of TAMs having a complex dual role, either oncogenic or anti-oncogenic, a “macrophage balance” theory has been proposed [27] to justify the complex dual role in their interactions with tumor cells. As an aside, although special attention has been paid to M2 macrophages here, M1 macrophages should not be overlooked. There were indications [28,29] that M1 macrophages seemed to eliminate their stereotyped impression as factors that act against tumors in the TME. In particular, in a recent study by You Y et al. [29], M1 macrophages were claimed to promote the malignant behaviors of oral squamous cell carcinoma cells, exemplified by in vitro evidence mainly collected from cell culture systems along with bioinformatic analyses. Nevertheless, the authors failed to consider the confirmation of M1 function in macrophages that they had already observed in an animal model, and the conclusion needs to be further examined. Notably, all studies that will be included in this review are mainly confined to those that explicitly study M2 macrophages in the cancer setting.

## 4. M2 Macrophages in the Tumor Microenvironment

Tumors are not only composed of malignant cells, but they can also contain a variety of cell types other than tumor cells. Macrophages are a major constituent, accounting for approximately 50% of nonmalignant cells [27,30] in certain types of cancer; the macrophage population consists not only of resident tissue macrophages, but also of differentiated monocytes that are recruited to the tumor from the blood. The first indication that cancer and inflammation are entangled with each other was provided by Virchow R in 1863 [30]. Although this suggestion has fallen out of favor over time, mounting evidence in recent years reiterates that inflammatory cells feature the tumor microenvironment, which supports the notion that they represent key players involved in the progression and development of tumors. Several lines of evidence [31,32,33] have strongly highlighted macrophages as the major players in cancer-related inflammation. For more information regarding macrophages engaged in cancer-related inflammation, refer to [31,34].

## 5. Prognostic Significance of M2 Macrophages in the Tumor Microenvironment

A strong correlation has been observed between an increased density of M2 macrophages and poor prognoses [35,36]. The suggestion that M2 macrophages promote tumors was made based on observations concerning the association between high frequencies of infiltrating M2 macrophages and inferior prognoses in many different human cancers, as is comprehensively reviewed elsewhere [35,37]. In accordance with these accounts, genes related to macrophage infiltration, such asCD68 or CD163 [38,39,40], were identified as part of the molecular signature that indicates dismal prognoses in patients with lymphoma. However, this concept has been challenged by a few reports showing that CD68 expressions related to macrophage infiltration have no relationship to the prognoses of patients with classic Hodgkin’s lymphoma [39,41] and liver cancer [42]. Therefore, it is controversial whether CD68 expressions can serve as a prognostic predictor. In the absence of additional evidence, it is difficult to conclude that CD68 can be used to indicate prognoses. In contrast, low levels of macrophage infiltration into tumors tend to be correlated with suppression of tumor growth and metastasis. Subsequent studies [43,44] have confirmed the positive association between M2 macrophage abundances in primary tumors and cancer metastasis in clinical sample tissues. Similar to what was determined from association studies utilizing clinical specimens, this tumor-promoting activity of M2 macrophages has also been underpinned by experiments using genetically modified mice [45] as well as in nude mice xenografted with tumor cells [46].

## 6. Performance of M2 Macrophages in the TME

As determined both from experimental and clinical investigations, the roles M2 macrophages play in the TME have become clear, which chiefly include metastasis and immune suppression. The behavior of M2 macrophages, of course, will be inescapably influenced or even regulated by the TME. Here, we will not provide an extended discussion from the perspective of cross-talk or interplay between the TME and behaviors of M2 macrophages, which is beyond the scope of our review. Instead, we will discuss the evidence that has been obtained regarding the involvement of M2 macrophages in promoting the metastasis and immunoevasion of tumor cells.

### 6.1. Propelling Metastasis

Much evidence for the metastasis-promoting role of M2 macrophages has come from in vitro cell co-culture systems and the use of elegant transgenic mouse models. By means of in vitro cell co-culture systems, the migration and invasion of tumor cells, usually exemplified by wound-healing and Transwell assays, were found to be strikingly promoted in the presence of M2 macrophages compared with the control. Case examples abound, regardless of tumor type. The molecular mechanism underlying how M2 macrophages foster metastasis of tumor cells can be summarized as follows: first, secreted proteins and non-coding RNAs, such as microRNAs [47] or long non-coding RNAs (lncRNAs), are involved [48]. M2 macrophages can use these non-coding RNAs to regulate the expressions of genes related to tumor cell metastasis; second, and similarly, released exosomes [49] contain non-coding RNAs or proteins related to metastasis, and M2 macrophages can promote the migration and invasion of tumor cells. These observations can also be achieved in vivo using transgenic mouse models [50].

### 6.2. Immunosuppressive Roles

Numerous influential studies [51,52,53] performed either in vitro or in vivo that employed single-cell sequencing or mapping techniques demonstrated that M2 macrophages residing in tumors have tumor-promoting and immune-suppressive traits. M2 macrophages, as stated before, can be induced and populate in response to IL-4, IL-10, and IL-13 [54]. Importantly, in terms of function, M2 macrophages counteract M1 macrophages. Not only do M2 macrophages aid tumor cells in avoiding host immune surveillance by secreting cytokines that are closely related to immune suppression, such as IL-10 and transforming growth factor beta (TGF-β), but M2 macrophages can also inhibit the differentiation of dendritic cells (DCs). Mechanistically, these cytokines produced by M2 macrophages can also curb the differentiation of DCs and inhibit this function by releasing IL-10 [55]. Furthermore, the molecular mechanism of immunosuppression mediated by M2 macrophages includes T-cell inactivation, along with the production of chemokines that preferentially recruit T-cell subpopulations without cytotoxic functions. However, this is not necessarily the case.M2 macrophages also express low levels of cytokines related to immune stimulation, such as IL-1, IL-12, and TNF [56]. In most cases, M2 macrophages are good producers of cytokines related to immunosuppression, giving rise to high levels of IL-10 and TGF-β [57]. These cytokines, especially TGF-β, can undoubtedly have a significant influence on the differentiation of regulatory T cells (Tregs), one of most important players involved in the TME that mainly mediates the suppression of cytotoxic T-cell immunity. Supporting this viewpoint, a line of research [58] from pattern analysis of gene expressions confirmed that, in addition to IL-10 and TGF-β, M2 macrophages were found to release high levels of CCL17, CCL18, and CCL22, signals that preferentially recruit Tregs and Th2 lymphocytes [59].

## 7. Different Types of PTMs That Occur in M2 Macrophages or TAMs

Having described both the definition and major actions of M2 macrophages in the TME, we next examined a variety of PTMs that have been reported to occur in M2 macrophages or extended to TAMs when little is known about PTMs in M2 macrophages. Regarded as an emerging research field, PTMs that have been reported to occur within M2 macrophages or related to TAMs include but are not limited to SUMOylation, methylation, lactylation, ubiquitylation, and acetylation. These PTMs more often than not actively participate in tilting toward M2 macrophages.

The different branches of a large tree can be used as an analogy to better comprehend the PTMs that take place in M2 macrophages (Figure 1). These PTMs may confer the functional plasticity and diversity of M2 macrophages in the pathophysiological setting of cancer. While this analogy offers a framework for understanding these PTMs that occur within M2 macrophages, this does not mean that each of the PTMs acts independently of each other. To the best of our knowledge, cross talk or interplay has been reported among these PTMs in certain physiological settings. Within the field of PTMs of M2 macrophages, a crucial question remains unanswered: whether there is a causal relationship between PTMs occurring within M2 macrophages and the TME is unknown and remains to be further established. The PTMs occurring within M2 macrophages, which are reviewed below, are in random order.

### 7.1. SUMOylation

Responding to different environmental factors, macrophages are highly plastic cells that are subject to classic polarization, namely M1 and M2 polarization, as stated above. Several lines of evidence have shown that SUMOylation is heavily involved in the polarization of M2 macrophages. A good example is the SUMOylation of Kruppel-like factor 4 (KLF4), a member of a subfamily of the zinc-finger class of DNA-binding transcription factors, which plays an essential role in M2 macrophage polarization [60]. The authors reported that SUMOylation of KLF4, induced by IL-4 treatment in macrophages, can promote RAW264.7 cells and bone-marrow-derived macrophages (BMDMs) to tilt toward the M2 subtype. A similar observation was also obtained from a recent article [61] regarding breast cancer research showing that increased Akt1 SUMOylation skewed macrophages toward M2 polarization within the breast cancer microenvironment, eventually promoting tumor progression. As of this writing, relevant studies concerning SUMOylation that occurs in M2 macrophages in the context of the TME have been scarce, other than the two studies reviewed above. In contrast, few studies have been published concerning SUMOylation occurring in macrophages; however, neither of these studies examined M2 macrophages or the TME. Given this, we will provide additional details here. More studies are required to further elucidate the function of SUMOylation in the polarization of M2 macrophages in the TME.

### 7.2. Methylation

Commonly seen in M2 macrophages aside from SUMOylation, methylation is also a type of PTM that has been found to occur within M2 macrophages. For example, Tikhanovich I et al. [62] discovered that protein arginine methyltransferase 1 (PRMT1) activities were seriously deficient in cirrhosis patients suffering from recurrent infections. Peroxisome proliferator-activated receptor gamma (PPARγ) is one of the key transcription factors that control macrophage polarization toward theM2 subtype. After treatment with rosiglitazone or GW1929, a commercial activator of PPARγ, M2 macrophage differentiation in vivo and in vitro was markedly restored. This pioneering investigation was important in that it first established a causal relationship between protein methylation and differentiation of M2 macrophages in an inflammation setting. Nevertheless, little is known regarding whether this causal relationship between methylation and M2 macrophage differentiation can be reproduced in M2 macrophages in the context of the TME in the absence of direct evidence that supports this hypothesis.

### 7.3. Lactylation

Originally identified by Zhang D et al. [63] in histones, lactylation has enriched the current knowledge regarding PTM types. Whether lactylation could take place in M2 macrophages in the TME remains largely unknown in view of sparse research available that explores the relationship between them. However, with the intention of reproducing the key findings made by Zhang D et al. [63], Dichtl S et al. [64] explored the lactylation of histones in M2 macrophages, and revealed that histone lactylation was dissociated from gene expressions pertaining to M2 macrophages, but was associated with arginase-1 expressions. Additionally, histone lactylation seems to be correlated with macrophage death under inflammatory stress. It should be noted that this investigation did not touch upon the role of histone lactylation in the polarization of M2 macrophages. Future studies are thus vital to investigate the correlations between histone lactylation and M2 macrophage polarization in the TME.

### 7.4. Ubiquitylation

Not only areM2 macrophages heavily engaged in tumor progression, they also take part in tissue homeostasis maintenance by scavenging dead cells, cell debris, and lipoprotein aggregates via their classic phagocytosis. With the aim of mechanistically determining the implications of M2 macrophages in tissue homeostasis, Guo M et al. [65] demonstrated, using a proteomics approach, that it was through polyubiquitylation of the macrophage scavenger receptor 1 (MSR1) at 63 arginine (K63) that recruits the TAK1/MKK7/JNK signaling complex to phagosomes of IL-4-activated M2 macrophages. From these data, the authors reasoned that polyubiquitylation of MSR1 at K63 may have implications for phenotypic switching between M1 and M2 macrophages in vivo in ovarian cancer patients. More research utilizing proteomics is needed to better understand the involvement of ubiquitylation in phenotypic switching of M2 macrophages in cancer.

### 7.5. Acetylation

Kolliniati et al. stressed the importance of metabolic shifts in the polarization of macrophages to distinct functional states [66]. Covarrubias AJ et al. [67] noted that the IL-4 signaling pathway collaborated with the Akt-mTORC1 pathway and was capable of regulating Acly, a key enzyme in Ac-CoA synthesis, contributing to increased histone acetylation and ultimately leading to M2 gene induction. The data provided convincing evidence of a link between acetylation and induction of M2 macrophages. The authors underscored the involvement of metabolism on histone acetylation in the polarization of M2 macrophages. Interestingly, a contrasting indication was provided in another separate study performed by Namgaladze D et al. [68], who established, using CRISPR/Cas9 technology, ATP-citrate lyase (ACLY) knockout human THP-1 macrophages where histone acetylation levels were reduced after knockout of ACLY, which is a key enzyme linked to the IL-4-induced polarization of murine bone marrow-derived macrophages.IL-4-induced gene expressions remained intact. Therefore, the role of histone acetylation in the polarization of M2 macrophages remains debatable and remains to be further examined.

### 7.6. Phosphorylation

To date, little attention has been paid to protein phosphorylation that occurs within M2 macrophages in the setting of the TME. However, few studies have investigated protein phosphorylation in tumor-associated macrophages (TAMs). Given these findings, the discussion we intended to provide here that resolves the phosphorylation that occurs within M2 macrophages must be extended from M2 macrophages to TAMs. By isolating TAMs from both human and murine tumor tissues, Su P et al. [69] discovered that TAMs were enriched with lipids owing to the increased lipid uptake by macrophages and that TAMs used fatty acid oxidation instead of glycolysis, which was preferred by tumor cells for energy provision. High levels of fatty acid oxidation can result inJAK1phosphorylation and SHP1dephosphorylation, thereby contributing to the activation of STAT6 and transcription of genes regulating TAM function and generation. This study was meaningful because it hinted at a link between the variations in protein phosphorylation and functions of TAMs. Consistent with this finding, a previous study by Tariq M et al. [70] reported that gefitinib, when used at low concentrations, can significantly suppress the polarization of M2 macrophages induced by IL-13 by inhibiting phosphorylation of STAT6, a key signaling pathway in the polarization of M2macrophages. All of this evidence, whether direct or indirect, explicitly suggested the involvement of phosphorylation in the polarization of M2 macrophages.

### 7.7. Glycosylation

Depending on the identity of the amino acid molecule that binds the carbohydrate chain, glycosylation can be classified into O-linked, N-linked, C-linked, or S-linked subtypes. Among them, only O-linked glycosylation, namely so-called O-GlcNAcylation, can take place on both intracellular and extracellular proteins [71]. Similarly, increasing evidence has revealed that O-GlcNAcylation is heavily involved in the polarization of M2 macrophages. For example, Rodrigues Mantuano N et al. [72] showed that increased glucose flow through the hexosamine biosynthetic pathway drove cancer progression and immune evasion by up-regulating O-GlcNAcylation in TAMs. Increased O-GlcNAcylation is skewed toward the M2 phenotype, favoring tumor progression. In line with this observation, Hinshaw DC et al. [73] demonstrated that, upon treatment with vismodegib, an inhibitor of the Hedgehog signaling pathway, O-GlcNAcylation was significantly decreased in TAMs isolated from murine mammary tumors. Decreased O-GlcNAcylation skews TAMs toward M1 macrophages. The two articles reviewed above strongly indicate that O-GlcNAcylation is required for the maintenance and regulation of M2 macrophage characteristics in the TME. With regards to the role of glycosylation in the polarization of TAMs, Mantuano NR et al. [74] performed a comprehensive and well-balanced review regarding glycosylation in TAMs. In this review, not only did the authors summarize the literature related to glycosylation occurring in TAMs, but they also included relevant studies concerning glycosylation taking place in pathophysiological contexts other than cancer. In contrast, in our current review, we focused on studies with respect to glycosylation that occurred only in TAMs or M2 macrophages in the TME, regardless of other disorders reported.

### 7.8. Neddylation

Analogous to ubiquitination, neddylation is a type of PTM that adds a ubiquitin-like protein, such asNEDD8, to substrate proteins [75]. Nedd is an ubiquitin-like protein whose neddylation hinges on E3 ligase. Thus, neddylation can lead to conformational changes in proteins and, consequently, may have an impact on protein–protein interactions. The published literature related to neddylation has revealed that neddylation regulates many important biological processes, including tumorigenesis [76], which means that neddylation is potentially targetable in the treatment of cancer [77]. The original report that established the relationship between neddylation and polarization of M2 macrophages was published by Asare Y and coworkers [78]. By utilizing MLN4924, an inhibitor of neddylation, the authors demonstrated that deneddylation with MLN4924 can skew macrophages to M2 macrophages, with classic biomarkers, including IL-13 and arginase-1, which was related to increases in M2 macrophages, whereas M1 macrophage biomarkers decreased. Similarly, global inactivation of neddylation by MLN4924 significantly exacerbated inflammation that was reminiscent of M1 macrophages. A case in point was the research conducted by Lin Y et al. [79] in a chronic pancreatitis context, showing that inactivation of neddylation by MLN4924 can increase CCL5 release and aggravate pancreatitis. In addition, for more information regarding the implications of neddylation in macrophages, please refer to the review by Jiang YY et al. [80] in which the authors conducted a comprehensive review of neddylation in macrophages.

### 7.9. Palmitoylation

Palmitoylation is an acylation modification in nature and is a reversible PTM where palmitate groups can be covalently attached to or detached from proteins [81,82]. Capable of altering the localization, protein stability, and functions of proteins in cells, palmitoylation is essential for the functions of both oncogenes and tumor-suppressing genes [83], hence having important implications for the pathophysiological activities occurring in cancer cells [84]. Furthermore, in light of such unique reversibility, palmitoylation allows proteins to be rapidly shuttled between biological membranes and cytoplasmic substrates in a process regulated by palmitoyl S-acyltransferases [84]. In a recent study that examined palmitoyl-protein thioesterase 1 (PPT1) in melanoma [85], Sharma G et al. discovered that, regardless of whether genetic or chemical inhibition of PPT1 mainly mediated deacetylation, it can lead to conversion of the M2 to the M1 phenotype in TAMs. To investigate the molecular mechanism underlying PPT inhibition leading to phenotypic switching of macrophages, the authors further found that such switching relies primarily on mitochondrial calcium release and p38 activation, along with induction of the cGAS/STING/TBK1 pathway, eventually synergizing with the antitumor activity of anti–PD-1 antibody in melanoma. From this study, it is conceivable that palmitoylation occurring in macrophages might enhance inflammation by skewing M2 to M1 phenotype switching. Since publication of this study, little has been published regarding palmitoylation occurring within M2 macrophages in the TME. Consequently, palmitoylation deserves more research attention in the future.

## 8. Technical Considerations when Studying M2 Macrophages in Tumors

In experimental analyses of M2 macrophages in tumors, some difficulties are unavoidable that could discourage investigators from conducting such analyses. For instance, the ideal condition for studying M2 macrophages in tumors was presumed to be in situ, rather than using transplantable models. Isolation of macrophages was subject to artifacts that were hard to discern, particularly when flow cytometric analysis was not coupled with immunocytochemistry in situ. Although highly similar in their genomes, transgenic mouse models that were used do not necessarily replicate human tumors, which are usually extensively studied at an advanced stage. In addition, the interplay between M2 macrophages and tumor cells can lead to novel gene expressions in both cell types; however, only partial data can in effect be reproduced in co-cultivation systems established in vitro that mimic in vivo interactions. In view of the technical limitations described above, the technical platforms employed in the research of M2 macrophages in tumors obviously need to be further refined to reveal new roles that have not been recognized.

## 9. Inhibitors or Drugs That Target PTMs

PTMs have attracted increasing attention in light of their importance and roles in cellular functions. PTMs can lead to the activation, inactivation, and complete alteration of protein functions, thus modulating a variety of molecular and cellular processes. For decades, both the number and types of PTMs have been determined and characterized, and these have drastically and rapidly increased. Thanks to advances in databases and bioinformatics, some PTMs that are expected to be detected based on conventional spectrometry have now been predicted by computer-based algorithms and machine learning [86,87,88]. Analyses of PTMs present a great opportunity for PTM-inspired drug design [89,90]. Generally, the majority of drug designs are based on the structure and chirality of kinases involved in the processes of PTMs. Few studies have explored the possibility of directly targeting the proteins in which PTMs occur, proving that the effect of selectively targeting a PTM could be better than directly targeting the entire kinase in charge of a PTM. A good example is provided by the study of Lisi S et al. [91]. The authors precisely interfered with the acetylated lysine 9 of histone H3 (H3K9ac) rather than with the entire histone acetylase transferase (HAT), and the interferential effect turned out to be more specific compared with the HAT inhibitors. This finding was important because it sheds new light on the paradigm shift from inhibition of the writer enzyme to acting on the PTM itself. Although an increasing number of PTMs have been characterized, many important kinases responsible for the occurrence of PTMs have not been determined. As a result, there has been little incentive to attempt to develop inhibitors. This probably mirrors the few literature studies that document the development of drugs or inhibitors that specifically target certain PTMs. The only literature report that we found targeted SUMOylation. Sumam de Oliveira D et al. [92], through a mini-review regarding SUMOylation, which is heavily engaged in regulating conserved biological processes in malaria parasites, highlighted the notion that targeting SUMOylation in *Plasmodium* is a potential target for malaria therapy. The study reported here deepens our understanding of PTMs in disease treatment. As of this writing, no relevant studies addressing PTMs in the cancer setting have been published, let alone in the TME.

## 10. Artificial Intelligence (AI) and TAMs

With the advent of artificial intelligence (AI), we willingly or unwillingly enter a new era of digital pathology. Capable of greatly assisting clinical pathologists in routine pathology diagnostics and of emancipating clinical pathologists from the overload of routine diagnostics [93,94], AI can also be applied to investigate the TME [95]. AI represents a state-of-the-art approach and has drawn considerable attention, clinically and basically, mainly because AI can identify or even discover new information regarding tumors that routine diagnostics may overlook [96,97]. To date, a number of studies have explored the advantages of AI relative to conventional methods, especially for analyzing new information when making diagnoses [98,99,100]. Considering that AI is a major technology that will cause our review to depart from the main theme, we discuss some reports that are closely related to analyses of TAMs. By examining the available related literature, only two studies were found. In one study, Cancian P et al. [101] attempted to employ a deep-learning pipeline to characterize TAMs in colorectal cancer liver metastasis. Through their comparisons, the authors found that AI can totally and successfully recognize TAMs embedded in the tumor microenvironment. In another study of diffuse large B-cell lymphoma (DLBCL), Carreras J et al. [102] used artificial neural networks to predict the overall prognosis and molecular subtypes of DLBCL, showing that prognoses and molecular subtypes can be predicted with high accuracy using neural networks. A review of these two reports suggests the promising application of AI in the analysis of TAMs in the cancer setting.

## 11. Conclusions

The purpose of this review was to summarize the past and present knowledge of M2 macrophages and their roles in the tumor microenvironment. The second aim of the review was to combine the literature closely related to PTMs that occur during the polarization of M2 macrophages in the context of cancer. To date, PTMs have been identified that have a marked influence on the differentiation and polarization of M2 macrophages, including but not limited to SUMOylation, methylation, lactylation, ubiquitylation, acetylation, phosphorylation, glycosylation, neddylation, and palmitoylation. Some of these modifications could favor skewing macrophages toward the M2 state, while others had opposite effects. A possible reason may be that, in response to different tissue microenvironments, M2 macrophages may adopt different modification strategies to fit into the tumor milieu. More new PTMs remain to be identified in the differentiation and polarization of M2 macrophages in future studies.

## Figures and Tables

**Figure 1 cancers-14-05532-f001:**
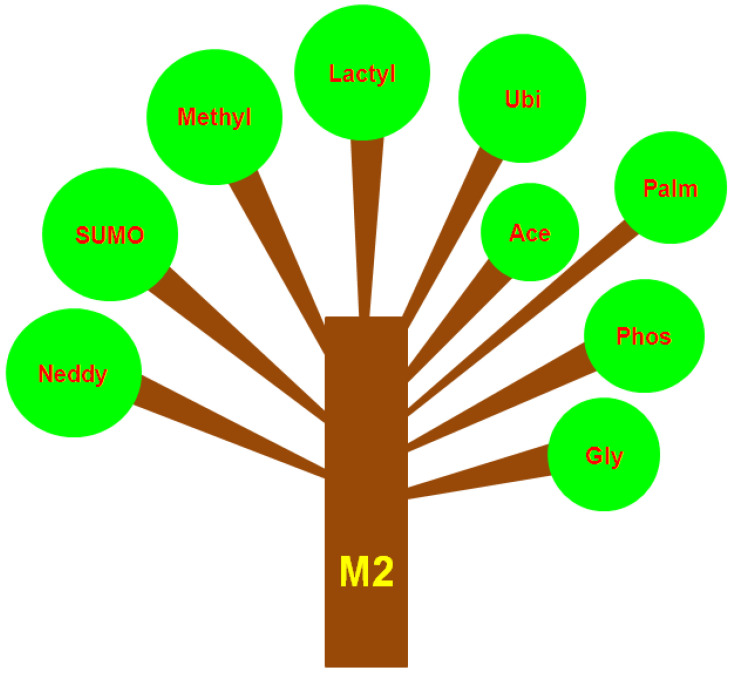
PTMs occurring in M2 macrophages can be likened to the different branches of a thick tree. These branches endow the tree with nutrition and energy from the environment that are necessary for its growth. Analogous to branches of the tree, PTMsare also important for M2 macrophages during their service in the tumor microenvironment, which endows functional diversity and plasticity for M2 macrophages. In these schematic graphics, the trunk-labeled M2 actually stands for M2 macrophages and the different branches represent the different PTMs, including neddylation (neddylation), SUMO (SUMOylation), methyl (methylation), lactyl (lactylation), Ubi (ubiquitylation), Ace (acetylation), Palm (palmitoylation), Phos (phosphorylation), and Gly (glycosylation).

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
