# Peer review of "Small Tweaks, Major Changes: Post-Translational Modifications That Occur within M2 Macrophages in the Tumor Microenvironment"

_cancers, 2022, doi:10.3390/cancers14225532_

Round 1

Reviewer 1 Report

This review has attempted to summarise different types of post-translational modifications (PTMs) in M2 macrophages in relation to the tumour microenvironment.  The quality of English has prevented its proper review as the majority of sentences contain multiple grammatical or syntax errors, with the repeated inversion of sentences at the start of the document making it very difficult to read. The authors could consider the following points.

1. English language corrections are essential prior to any subsequent review of this manuscript. English language errors are so widespread that it is likely that English language editing may generate statements that are false, necessitating review of the manuscript after its amendment. A professional English language editorial service should be used.

2. With regards to Figure 1, I understand that the authors are trying to depict that different PTMs each contribute to the M2 macrophage phenotype, however, the manner that this figure is drawn seems to be quite misleading. Firstly, the trunk of a tree (depicted as the M2 macrophage) gives rise to the branches (in Figure 1 depicted as the PTMs). This is at odds with the scenario that the figure is meant to depict, that the PTMs are what drive the generation of the M2 macrophage. Secondly, the manner that the diagram is drawn suggests that each of the PTMs acts independently of each other, which we know not to be the case. There does not seem to be a particular need for a diagram in this review, however if the authors are keen to include one, I suggest that they reconsider its structure and amend it so that it more accurate depicts the topic summary.

3. At the end of the article, the authors correctly state that the experimental systems used to generate much of the data that they have presented is unlikely to represent the tumour microenvironment. However, it is not clear which of the findings that are summarised in the body of the manuscript were generated only from artificial in vitro culture models and which were performed in or supported by inclusion of in vivo data. As this information is pivotal for understanding the robustness of results, inclusion of some details would improve the usefulness of the review. Due to the many shortfalls in this area of research, it may be necessary to only indicate where data were derived from or supported by in vivo studies.

4. Suggestion: A number of approved drugs and drugs under development target PTMs, largely based on the known functions of these PTMs in driving the growth and progression of particular cancers. Potential effects of these classes of drugs on M1/M2 polarisation and downstream effects on anti-tumour immune reactions could be better presented.

Author Response

Comments and Suggestions for Authors

This review has attempted to summarise different types of post-translational modifications (PTMs) in M2 macrophages in relation to the tumour microenvironment.  The quality of English has prevented its proper review as the majority of sentences contain multiple grammatical or syntax errors, with the repeated inversion of sentences at the start of the document making it very difficult to read. The authors could consider the following points.

  1. English language corrections are essential prior to any subsequent review of this manuscript. English language errors are so widespread that it is likely that English language editing may generate statements that are false, necessitating review of the manuscript after its amendment. A professional English language editorial service should be used.

Response: Agreeing with the reviewer, we have had our paper polished by AJE (American Journal of Experts) company. All the textual errors, we believe, have been weeded out with the aid of AJE. It should be noted that, our revised version used track changes, to differentiate from the old version.

  1. With regards to Figure 1, I understand that the authors are trying to depict that different PTMs each contribute to the M2 macrophage phenotype, however, the manner that this figure is drawn seems to be quite misleading. Firstly, the trunk of a tree (depicted as the M2 macrophage) gives rise to the branches (in Figure 1 depicted as the PTMs). This is at odds with the scenario that the figure is meant to depict, that the PTMs are what drive the generation of the M2 macrophage. Secondly, the manner that the diagram is drawn suggests that each of the PTMs acts independently of each other, which we know not to be the case. There does not seem to be a particular need for a diagram in this review, however if the authors are keen to include one, I suggest that they reconsider its structure and amend it so that it more accurate depicts the topic summary.

Response: with all due respect, we see it differently. We compared the trunk to M2 macrophages, the different branches spreading from trunk were likened to different post-translational modifications (PTMs) that occurred within M2 macrophages. It is these branches that energized the tree. By the same token, it is these PTMs that confer the functional diversity to M2 macrophages. We don’t think our figure can be misleading. The figure 1, in a sense, can help readers better understand the physiological significance of PTMs. So, we are keen to include this figure.

On the other hand, the reviewer’s misgiving that figure is drawn seems to be quite misleading, which is understandable. To drive away this misgiving, we have stressed that “while this analogy offers a framework for understanding these PTMs occurring within M2 macrophages, it does not mean that each of the PTMs acts independently of each other. As far as we know, there existed cross-talk or interplay reported among these PTMs in certain physiological settings. Within the field of PTMs of M2 macrophages, a crucial question remains unanswered is that, whether there existed the causal relationship between PTMs occurring within M2 macrophages and TME was unknown which is left to be further established”, which was highlighted to differentiate from the old contents. We hope that the reviewer would understand and be satisfied with what we revised.

  1. At the end of the article, the authors correctly state that the experimental systems used to generate much of the data that they have presented is unlikely to represent the tumour microenvironment. However, it is not clear which of the findings that are summarised in the body of the manuscript were generated only from artificial in vitro culture models and which were performed in or supported by inclusion of in vivo data. As this information is pivotal for understanding the robustness of results, inclusion of some details would improve the usefulness of the review. Due to the many shortfalls in this area of research, it may be necessary to only indicate where data were derived from or supported by in vivo studies.

Response: we appreciate the reviewer’s suggestion that is pertinent and constructive. Yes, when paraphrasing or introducing the previous works that were closely related to ours, we have, to the best of our abilities, included the details of experiments undertaken, in vitro or in vivo, in these studies we have included in our review, as suggested.

  1. Suggestion: A number of approved drugs and drugs under development target PTMs, largely based on the known functions of these PTMs in driving the growth and progression of particular cancers. Potential effects of these classes of drugs on M1/M2 polarization and downstream effects on anti-tumour immune reactions could be better presented.

Response: we strongly agree with suggestion given by the reviewer. We have included a separate section subtitled Inhibitors or drugs that target PTMs

PTMs have attracted increasing attention in light of their importance and roles in cellular function. PTM can lead to activation, inactivation and complete alteration of protein function therefore modulating a variety of molecular and cellular processes. For decades, both the number and types of PTMs have been found and characterized have drastically and rapidly increased. Thanks to the advance of database and bioinformatics, some PTMs that were supposed to be detected based on conventional spectrometry have now been predicted by computer-based algorithms and machine learning [86-88]. Analyses of PTMs present a great opportunity towards PTM-inspired drug design [89, 90]. Generally, the majority of drug design was usually based on the structure and chirality of kinases involved in the process of PTMs. Few study explored the possibility that directly targets the protein to which PTM occurs, proving that the effect of selectively targeting the PTM could be better than directly targeting the whole kinase in charge of PTM. A good example is from the study made by Lisi S et al [91]. The authors through precisely interfering the acetylated lysine 9 of histone H3 (H3K9ac), rather than the whole histone acetylase transferase (HAT), the interferential effect turned out to be more specific compared with the HAT inhibitors. This finding was important in that it sheds new light on paradigm shifting from inhibition of the writer enzyme to acting on the PTM itself. Despite increasingly more and more PTMs were characterized, during which many important kinases responsible for occurrence of PTMs have been far from determined. as a result there has been little incentive to seek to develop inhibitors. This probably mirrors the a few body of literatures documenting the development of drug or inhibitor specifically targeting certain PTMs. The only one literature we searched out is from targeting SUMOylation. Sumam de Oliveira D et al [92], through mini-review regarding SUMOylation that is heavily engaged in the regulation of conserved biological processes in malaria parasites, highlighted the notion that targeting SUMOylation in Plasmodium as a potential target for malaria therapy. The study reported here deepens our understanding about the PTM in the curing of disease. As of this writing, no relevant study looking into PTM has come up in cancer setting, let alone in TME. This separate section was also hightlighted.

Reviewer 2 Report

Zheng et al. aimed to summarize the post-translational modifications (PTM) that occur in M2 macrophages that play a pivotal role in the tumor microenvironment in this review work. The authors showed that the information for the works of literature published concerning M2 macrophages but unmentioned with the PTMs. Furthermore, including some of these modifications that could favor skewing macrophages toward the M2 state was also discussed in this manuscript. While this is an exciting direction to explore with research, there are still some issues that need to be addressed:

1.          The authors refer to the post-translational modifications (PTM) as a mediator that occurred in M2 macrophages in the tumor microenvironment (TME). However, the manuscript discusses the PTM more than the mechanisms or pathways of M2 macrophage effects. In addition, as a mediator, the changes in TME in the body can also be a tool for cancer diagnosis. The manuscript should also be proposed in the manuscript to understand more clearly the influence of PTM on the M2 macrophages in TEM.

2.          In addition to the several modifications and modalities mentioned by the authors, can more research modalities be used as candidates to calculate the effect of M2 macrophages? Nowadays, artificial intelligence data analysis processing is more convenient. Can the database be used as a simulation of the calculation process for analytical diagnostics, evaluation of modification, or novel pathway detection? All these should be proposed in the manuscript.

3.          The authors list several mechanisms in Table 1. Do they have a corresponding pathway to regulate the mechanism? It can be organized in a table for precise classification if relevant information is available.

4.          The mechanistic analyses need to involve the performance of the PTM concerning the state of M2 macrophages in the TME. Does this mean that if the PTM is analyzed, different TMEs can be identified? If so, the M2 macrophages modified by different PTM can be classified to confirm whether there is a correlation with the mechanism of cancer generation.

5.          The authors could have compiled and presented several particularly important articles and their experimental results to make the manuscript more convincing.

Author Response

Comments and Suggestions for Authors

Zheng et al. aimed to summarize the post-translational modifications (PTM) that occur in M2 macrophages that play a pivotal role in the tumor microenvironment in this review work. The authors showed that the information for the works of literature published concerning M2 macrophages but unmentioned with the PTMs. Furthermore, including some of these modifications that could favor skewing macrophages toward the M2 state was also discussed in this manuscript. While this is an exciting direction to explore with research, there are still some issues that need to be addressed:

Response: we appreciate the reviewer so much for his/her taking time to patiently and critically go through our review paper, and for the commenting that “this is an exciting direction to explore with research”.

  1. The authors refer to the post-translational modifications (PTM) as a mediator that occurred in M2 macrophages in the tumor microenvironment (TME). However, the manuscript discusses the PTM more than the mechanisms or pathways of M2 macrophage effects. In addition, as a mediator, the changes in TME in the body can also be a tool for cancer diagnosis. The manuscript should also be proposed in the manuscript to understand more clearly the influence of PTM on the M2 macrophages in TEM.

Response: yes, we agree with the reviewer. Actually, PTM has been our main theme with which we have been concerned in this paper, which deserves more attention than other content. Despite the relevant studies regarding PTM in M2 macrophage were not much, even scant; review of these studies available reveals that the majority of these studies we included were shallow that lacks of mechanistic insight that how PTMs influenced M2 macrophages in TME. So, given this, we have to reduce the descriptions regarding the mechanism. Hope the reviewer could understand.

  1. In addition to the several modifications and modalities mentioned by the authors, can more research modalities be used as candidates to calculate the effect of M2 macrophages? Nowadays, artificial intelligence data analysis processing is more convenient. Can the database be used as a simulation of the calculation process for analytical diagnostics, evaluation of modification, or novel pathway detection? All these should be proposed in the manuscript.

Response: yes, this is a constructive suggestion that we totally agree. With regard to the application of artificial intelligence in the prediction of PTMs and in the analysis of TME, which have been included at the end of our review paper at your suggestion. For your convenience, the content newly added was posted below:

Artificial intelligence (AI) and TAMs.

With the advent of artificial intelligence (AI), willingly or unwillingly, we enter into a new era about digital pathology. Capable of greatly assisting the clinical pathologists in routine pathology diagnostics and of emancipating the clinical pathologists from overload of routine diagnostics [93, 94], AI can be applied to investigate the TME as well [95]. Actually, representing the state of the art approach, AI has drawn considerable attention, clinically and basically, mainly in that it can identify or even discover the new information of tumors that routine diagnostics may overlook [96, 97]. So far, a number of studies have explored the advantage that is incomparable to conventional methods, especially in analyzing the new information when diagnosing [98-100]. Considering that AI is a big story that setting out it will make our review deviate from the main theme, here we have to just cover some reports closely surrounding the analyses of TAMs. Combed through the available literature that is related, only two studies happened to be searched out. In one piece of study, Cancian P et al [101] attempted to employ deep-learning pipeline to characterize the TAMs in colorectal cancer liver metastasis. Through comparisons, the authors found that AI can totally and successfully recognize the TAMs that embedded in tumor microenvironment. In another study from diffuse large B-cell lymphoma (DLBCL), Carreras J et al [102] took advantage of artificial neural networks to predict the overall prognosis and molecular subtypes of DLBCL, showing that prognoses and molecular subtypes can be predicted with high accuracy using neural networks. A review from these two reports suggested the promising application of AI in the analysis of TAMs in cancer setting.

  1. The authors list several mechanisms in Table 1. Do they have a corresponding pathway to regulate the mechanism? It can be organized in a table for precise classification if relevant information is available.

Response: with all due respect, there was no table at all throughout our review. Signaling pathway regulating the mechanism in tumor microenvironment will be another big story that telling of it undoubtedly beyond the scope of our review. In addition, covering and introduction of the signaling pathway involved could make our review seem to be loosely interwoven. Given these, we will not cover the relevant signaling pathway that got involved here; instead, we just focused on the main theme—the PTMs that occurred within the M2 macrophages in TME.

4.The mechanistic analyses need to involve the performance of the PTM concerning the state of M2 macrophages in the TME. Does this mean that if the PTM is analyzed, different TMEs can be identified? If so, the M2 macrophages modified by different PTM can be classified to confirm whether there is a correlation with the mechanism of cancer generation.

Response: with all due respect, we see it differently. Whether there will exist causal relationship between occurrence of PTM and different TMEs, remains unknown. At least, we have not found there were related documents supporting it.

  1. The authors could have compiled and presented several particularly important articles and their experimental results to make the manuscript more convincing.

Response: we agree with the review on this point. As a matter of a fact, each landmark study, or seminal/influential/pioneering study has to be incited and covered as much as we can where appropriate to support our argument.

Round 2

Reviewer 1 Report

While some of the English language errors have been corrected in this manuscript, numerous grammatical and syntax errors remain. In addition, many inverted sentences are awkward to read and difficult to understand, the use of conversational or colloquial English is inappropriate for a scientific article, the interchangeable use of present and past tenses is incorrect in places and many errors have not been corrected. Repetition of sentences throughout the manuscript could also be removed by efficient English language editing. I hope that it will be possible for the authors to explain the types of corrections and amendments that they require to an English language editor who has scientific expertise.

Author Response

Respond: we have turned to English language service provided by AJE company once again asking further polishing and refinement, hoping this time all that polished could be satisfying.

Reviewer 2 Report

The authors have resolved most of the concerns proposed by the reviewer, and the manuscript has been improved significantly. Therefore, we do not have further revision requirements for this updated manuscript.

Author Response

Respond: we appreciate the reviewer so much for his or her taking time and patience to check our revised manuscript and for stating that the authors have resolved most of the concerns proposed by the reviewer, and the manuscript has been improved significantly.
